# COLORING GRAPH NEURAL NETWORKS FOR NODE DISAMBIGUATION

## ABSTRACT

In this paper, we show that a simple coloring scheme can improve, both theoretically and empirically, the expressive power of Message Passing Neural Networks (MPNNs). More specifically, we introduce a graph neural network called Colored Local Iterative Procedure (CLIP) that uses colors to disambiguate identical node attributes, and show that this representation is a *universal approximator* of continuous functions on graphs with node attributes. Our method relies on *separability*, a key topological characteristic that allows to extend well-chosen neural networks into universal representations. Finally, we show experimentally that CLIP is capable of capturing structural characteristics that traditional MPNNs fail to distinguish, while being state-of-the-art on benchmark graph classification datasets.

## 1 INTRODUCTION

Learning good representations is seen by many machine learning researchers as the main reason behind the tremendous successes of the field in recent years (Bengio et al., 2013). In image analysis (Krizhevsky et al., 2012), natural language processing (Vaswani et al., 2017) or reinforcement learning (Mnih et al., 2015), groundbreaking results rely on efficient and flexible deep learning architectures that are capable of transforming a complex input into a simple vector while retaining most of its valuable features. The *universal approximation theorem* (Cybenko, 1989; Hornik et al., 1989; Hornik, 1991; Pinkus, 1999) provides a theoretical framework to analyze the expressive power of such architectures by proving that, under mild hypotheses, multi-layer perceptrons (MLPs) can uniformly approximate any continuous function on a compact set. This result provided a first theoretical justification of the strong approximation capabilities of neural networks, and was the starting point of more refined analyses providing valuable insights into the generalization capabilities of these architectures (Baum and Haussler, 1989; Geman et al., 1992; Saxe et al., 2014; Bartlett et al., 2018).

Despite a large literature and state-of-the-art performance on benchmark graph classification datasets, graph neural networks yet lack a similar theoretical foundation (Xu et al., 2019). Universality for these architectures is either hinted at via equivalence with approximate graph isomorphism tests ($k$-WL tests in Xu et al. 2019; Maron et al. 2019a), or proved under restrictive assumptions (finite node attribute space in Murphy et al. 2019). In this paper, we introduce Colored Local Iterative Procedure[1] (CLIP), which tackles the limitations of current Message Passing Neural Networks (MPNNs) by showing, both theoretically and experimentally, that adding a simple coloring scheme can improve the flexibility and power of these graph representations. More specifically, our contributions are: 1) we provide a precise mathematical definition for universal graph representations, 2) we present a general mechanism to design universal neural networks using separability, 3) we propose a novel node coloring scheme leading to CLIP, the first provably universal extension of MPNNs, 4) we show that CLIP achieves state of the art results on benchmark datasets while significantly outperforming traditional MPNNs as well as recent methods on graph property testing.

The rest of the paper is organized as follows: Section 2 gives an overview of the graph representation literature and related works. Section 3 provides a precise definition for universal representations, as well as a generic method to design them using *separable* neural networks. In Section 4, we show that most state-of-the-art representations are not sufficiently expressive to be universal. Then, using the analysis of Section 3, Section 5 provides CLIP, a provably universal extension of MPNNs. Finally,

---

[1]Code will be available at https://github.com/ after the review process.

Section 6 shows that CLIP achieves state-of-the-art accuracies on benchmark graph classification taks, as well as outperforming its competitors on graph property testing problems.

## 2 RELATED WORKS

The first works investigating the use of neural networks for graphs used recurrent neural networks to represent directed acyclic graphs (Sperduti and Starita, 1997; Frasconi et al., 1998). More generic graph neural networks were later introduced by Gori et al. (2005); Scarselli et al. (2009), and may be divided into two categories. 1) *Spectral methods* (Bruna et al., 2014; Henaff et al., 2015; Defferrard et al., 2016; Kipf and Welling, 2017) that perform convolution on the Fourier domain of the graph through the spectral decomposition of the graph Laplacian. 2) *Message passing neural networks* (Gilmer et al., 2017), sometimes simply referred to as *graph neural networks*, that are based on the aggregation of neighborhood information through a local iterative process. This category contains most state-of-the-art graph representation methods such as (Duvenaud et al., 2015; Grover and Leskovec, 2016; Lei et al., 2017; Ying et al., 2018; Verma and Zhang, 2019), DeepWalk (Perozzi et al., 2014), graph attention networks (Velickovic et al., 2018), graphSAGE (Hamilton et al., 2017) or GIN (Xu et al., 2019).

Recently, (Xu et al., 2019) showed that MPNNs were, at most, as expressive as the Weisfeiler-Lehman (WL) test for graph isomorphism (Weisfeiler and Lehman, 1968). This suprising result led to several works proposing MPNN extensions to improve their expressivity, and ultimately tend towards *universality* (Maron et al., 2019a;b;c; Murphy et al., 2019; Chen et al., 2019). However, these graph representations are either as powerful as the $k$-WL test (Maron et al., 2019a), or provide universal graph representations under the restrictive assumption of finite node attribute space (Murphy et al., 2019). Other recent approaches (Maron et al., 2019c) implies quadratic order of tensors in the size of the considered graphs. Some more powerfull GNNs are studied and benchmarked on real classical datasets and on graph property testing (Kriege et al., 2018; Murphy et al., 2019; Chen et al., 2019): a set of problems that classical MPNNs cannot handle. Our work thus provides a more general and powerful result of universality, matching the original definition of (Cybenko, 1989) for MLPs.

## 3 UNIVERSAL REPRESENTATIONS VIA SEPARABILITY

In this section we present the theoretical tools used to design our universal graph representation. More specifically, we show that *separable* representations are sufficiently flexible to capture all relevant information about a given object, and may be extended into universal representations.

### 3.1 NOTATIONS AND BASIC ASSUMPTIONS

Let $\mathcal{X}, \mathcal{Y}$ be two topological spaces, then $\mathcal{F}(\mathcal{X}, \mathcal{Y})$ (resp. $\mathcal{C}(\mathcal{X}, \mathcal{Y})$) denotes the space of all functions (resp. continuous functions) from $\mathcal{X}$ to $\mathcal{Y}$. Moreover, for any group $G$ acting on a set $\mathcal{X}$, $\mathcal{X}/G$ denotes the set of orbits of $\mathcal{X}$ under the action of $G$ (see Appendix B for more details). Finally, $\|\cdot\|$ is a norm on $\mathbb{R}^d$, and $\mathcal{P}_n$ is the set of all permutation matrices of size $n$. In what follows, we assume that all the considered topological spaces are *Hausdorff* (see e.g. (Bourbaki, 1998) for an in-depth review): each pair of distinct points can be separated by two disjoint open sets. This assumption is rather weak (e.g. all metric spaces are Hausdorff) and is verified by most topological spaces commonly encountered in the field of machine learning.

### 3.2 UNIVERSAL REPRESENTATIONS

Let $\mathcal{X}$ be a set of objects (e.g. vectors, images, graphs, or temporal data) to be used as input information for a machine learning task (e.g. classification, regression or clustering). In what follows, we denote as *vector representation* of $\mathcal{X}$ a function $f : \mathcal{X} \to \mathbb{R}^d$ that maps each element $x \in \mathcal{X}$ to a $d$-dimensional vector $f(x) \in \mathbb{R}^d$. A standard setting for supervised representation learning is to define a class of vector representations $\mathfrak{F}_d \subset \mathcal{F}(\mathcal{X}, \mathbb{R}^d)$ (e.g. convolutional neural networks for images) and use the target values (e.g. image classes) to learn a *good* vector representation in light of the supervised learning task (i.e. one vector representation $f \in \mathfrak{F}_d$ that leads to a good accuracy on the learning task). In order to present more general results, we will consider neural network architectures that can output vectors of any size, i.e. $\mathfrak{F} \subset \cup_{d \in \mathbb{N}^*} \mathcal{F}(\mathcal{X}, \mathbb{R}^d)$, and will denote $\mathfrak{F}_d = \mathfrak{F} \cap \mathcal{F}(\mathcal{X}, \mathbb{R}^d)$

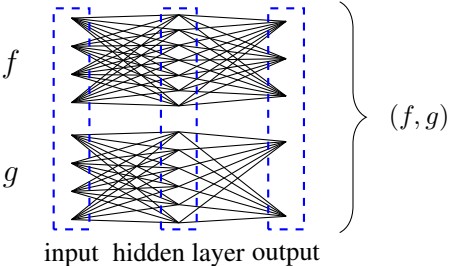
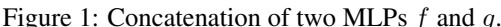

input  hidden layer output

Figure 1: Concatenation of two MLPs $f$ and $g$.

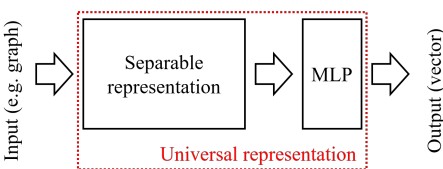

Figure 2: Universal representations can easily be created by combining a separable representation with an MLP.

the set of $d$-dimensional vector representations of $\mathfrak{F}$. A natural characteristic to ask from the class $\mathfrak{F}$ is to be generic enough to approximate any vector representation, a notion that we will denote as *universal representation* (Hornik et al., 1989).

**Definition 1.** A class of vector representations $\mathfrak{F} \subset \cup_{d \in \mathbb{N}^*} \mathcal{F}(\mathcal{X}, \mathbb{R}^d)$ is called a *universal representation* of $\mathcal{X}$ if for any compact subset $K \subset \mathcal{X}$ and $d \in \mathbb{N}^*$, $\mathcal{F}$ is uniformly dense in $\mathcal{C}(K, \mathbb{R}^d)$.

In other words, $\mathfrak{F}$ is a universal representation of a normed space $\mathcal{X}$ if and only if, for any continuous function $\phi : \mathcal{X} \to \mathbb{R}^d$, any compact $K \subset \mathcal{X}$ and any $\varepsilon > 0$, there exists $f \in \mathfrak{F}$ such that

$$\forall x \in K, \ \|\phi(x) - f(x)\| \le \varepsilon \,. \tag{1}$$

One of the most fundamental theorems of neural network theory states that one hidden layer MLPs are universal representations of the $m$-dimensional vector space $\mathbb{R}^m$.

**Theorem 1** (Pinkus, 1999, Theorem 3.1). *Let $\varphi : \mathbb{R} \to \mathbb{R}$ be a continuous non polynomial activation function. For any compact $K \subset \mathbb{R}^m$ and $d \in \mathbb{N}^*$, two layers neural networks with activation $\varphi$ are uniformly dense in the set $\mathcal{C}(K, \mathbb{R}^d)$.*

However, for graphs and structured objects, universal representations are hard to obtain due to their complex structure and invariance to a group of transformations (e.g. permutations of the node labels). We show in this paper that a key topological property, *separability*, may lead to universal representations of those structures.

### 3.3 SEPARABILITY IS (ALMOST) ALL YOU NEED

Loosely speaking, universal representations can approximate any vector-valued function. It is thus natural to require that these representations are *expressive* enough to separate each pair of dissimilar elements of $\mathcal{X}$.

**Definition 2** (Separability). A set of functions $\mathfrak{F} \subset \mathcal{F}(\mathcal{X}, \mathcal{Y})$ is said to *separate* points of $\mathcal{X}$ if for every pair of distinct points $x$ and $y$, there exists $f \in \mathfrak{F}$ such that $f(x) \neq f(y)$.

For a class of vector representations $\mathfrak{F} \subset \cup_{d \in \mathbb{N}^*} \mathcal{F}(\mathcal{X}, \mathbb{R}^d)$, we will say that $\mathfrak{F}$ is *separable* if its 1-dimensional representations $\mathfrak{F}_1$ separates points of $\mathcal{X}$. Separability is rather weak, as we only require the existence of different outputs for every pair of inputs. Unsurprisingly, we now show that it is a necessary condition for universality (see Appendix A for all the detailed proofs).

**Proposition 1.** *Let $\mathfrak{F}$ be a universal representation of $\mathcal{X}$, then $\mathfrak{F}_1$ separates points of $\mathcal{X}$.*

While separability is necessary for universal representations, it is also key to designing neural network architectures that can be extended into universal representations. More specifically, under technical assumptions, separable representations can be composed with a universal representation of $\mathbb{R}^d$ (such as MLPs) to become universal.

**Theorem 2.** *For all $d \ge 0$, let $\mathcal{M}_d$ be a universal approximation of $\mathbb{R}^d$. Let $\mathfrak{F}$ be a class of vector representations of $\mathcal{X}$ such that:*

*(i) **Continuity:** every $f \in \mathfrak{F}$ is continuous,*

*(ii) **Stability by concatenation:** for all $f, g \in \mathfrak{F}$, $x \mapsto (f(x), g(x)) \in \mathfrak{F}$,*

*(iii)* ***Separability:*** $\mathfrak{F}_1$ *separates points of* $\mathcal{X}$.

*Then* $\{\psi \circ f \; : \; \exists d \geq 1 \text{ s.t. } \psi \in \mathcal{M}_d, f \in \mathfrak{F}\}$ *is a universal representation of* $\mathcal{X}$.

Stability by concatenation is verified by most neural networks architectures, as illustrated for MLPs in Figure 1. The proof of Theorem 2 relies on the Stone-Weierstrass theorem (see e.g. Rudin, 1987, Theorem 7.32) whose assumptions are continuity, separability, and the fact that the class of functions is an algebra. Fortunately, composing a separable and concatenable representation with a universal representation automatically leads to an algebra, and thus the applicability of the Stone-Weierstrass theorem and the desired result. A complete derivation is available in Appendix A. Since MLPs are universal representations of $\mathbb{R}^d$, Theorem 2 implies a convenient way to design universal representations of more complex object spaces: create a separable representation and compose it with a simple MLP (see Figure 2).

**Corollary 1.** *A continuous, concatenable and separable representation of* $\mathcal{X}$ *composed with an MLP is universal.*

Note that many neural networks of the deep learning literature have this two steps structure, including classical image CNNs such as AlexNet (Krizhevsky et al., 2012) or Inception (Szegedy et al., 2016). In this paper, we use Corollary 1 to design universal graph and neighborhood representations, although the method is much more generic and may be applied to other objects.

## 4 LIMITATIONS OF EXISTING REPRESENTATIONS

In this section, we first provide a proper definition for graphs with node attributes, and then show that message passing neural networks are not sufficiently expressive to be universal.

### 4.1 GRAPHS WITH NODE ATTRIBUTES

Consider a dataset of $n$ interacting objects (e.g. users of a social network) in which each object $i \in [\![1, n]\!]$ has a vector attribute $v_i \in \mathbb{R}^m$ and is a node in an undirected graph $G$ with adjacency matrix $A \in \mathbb{R}^{n \times n}$.

**Definition 3.** The space of graphs of size $n$ with $m$-dimensional node attributes is the quotient space

$$\mathbf{Graph}_{m,n} = \left\{ (v, A) \in \mathbb{R}^{n \times m} \times \mathbb{R}^{n \times n} \right\} / \mathcal{P}_n \,, \tag{2}$$

where $A$ is the adjacency matrix of the graph, $v$ contains the $m$-dimensional representation of each node in the graph and the set of permutations matrices $\mathcal{P}_n$ is acting on $(v, A)$ by

$$\forall P \in \mathcal{P}_n, \quad P \cdot (v, A) = (Pv, PAP^\top) \,. \tag{3}$$

Moreover, we limit ourselves to graphs of maximum size $n_{\max}$, where $n_{\max}$ is a large integer. This allows us to consider functions on graphs of different sizes without obtaining infinite dimensional spaces and infinitely complex functions that would be impossible to learn via a finite number of samples. We thus define $\mathbf{Graph}_m = \bigcup_{n \leq n_{\max}} \mathbf{Graph}_{m,n}$. More details on the technical topological aspects of the definition are available in Appendix B, as well as a proof that $\mathbf{Graph}_m$ is Hausdorff.

### 4.2 MESSAGE PASSING NEURAL NETWORKS

A common method for designing graph representations is to rely on local iterative procedures. Following the notations of Xu et al. (2019), a *message passing neural network* (MPNN) (Gilmer et al., 2017) is made of three consecutive phases that will create intermediate node representations $x_{i,t}$ for each node $i \in [\![1, n]\!]$ and a final graph representation $x_G$ as described by the following procedure: 1) **Initialization:** All node representations are initialized with their node attributes $x_{i,0} = v_i$. 2) **Aggregation and combination:** $T$ local iterative steps are performed in order to capture larger and larger structural characteristics of the graph. 3) **Readout:** This step combines all final node representations into a single graph representation: $x_G = \text{READOUT}(\{x_{i,T}\}_{i \in [\![1,n]\!]})$, where READOUT is permutation invariant.

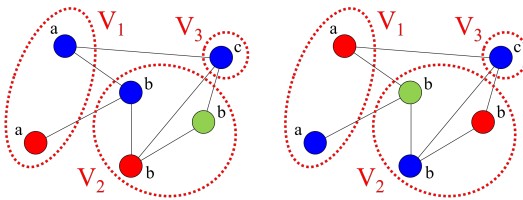

Figure 3: Example of two valid colorings of the same attributed graph. Note that each $V_k$ contains nodes with identical attributes.

Unfortunately, while MPNNs are very efficient in practice and proven to be as expressive as the Weisfeiler-Lehman algorithm (Weisfeiler and Lehman, 1968; Xu et al., 2019), they are not sufficiently expressive to construct isomorphism tests or separate all graphs (for example, consider $k$-regular graphs without node attributes, for which a small calculation shows that any MPNN representation will only depend on the number of nodes and degree $k$ (Xu et al., 2019)). As a direct application of Proposition 1, MPNNs are thus not expressive enough to create universal representations.

## 5 EXTENDING MPNNS USING A SIMPLE COLORING SCHEME

In this section, we present Colored Local Iterative Procedure (CLIP), an extension of MPNNs using colors to differentiate identical node attributes, that is able to capture more complex structural graph characteristics than traditional MPNNs. This is proved theoretically through a universal approximation theorem in Section 5.3 and experimentally in Section 6. CLIP is based on three consecutive steps: 1) graphs are colored with several different colorings, 2) a neighborhood aggregation scheme provides a vector representation for each colored graph, 3) all vector representations are combined to provide a final output vector. We now provide more information on the coloring scheme.

### 5.1 COLORS TO DIFFERENTIATE NODES

In order to distinguish non-isomorphic graphs, our approach consists in coloring nodes of the graph with identical attributes. This idea is inspired by classical graph isomorphism algorithms that use colors to distinguish nodes (McKay, 1981), and may be viewed as an extension of one-hot encodings used for graphs without node attributes (Xu et al., 2019).

For any $k \in \mathbb{N}$, let $C_k$ be a finite set of $k$ colors. These colors may be represented as one-hot encodings ($C_k$ is the natural basis of $\mathbb{R}^k$) or more generally any finite set of $k$ elements. At initialization, we first partition the nodes into groups of identical attributes $V_1, ..., V_K \subset [\![1, n]\!]$. Then, for a subset $V_k$ of size $|V_k|$, we give to each of its nodes a distinct color from $C_k$ (hence a subset of size $|V_k|$). For example, Figure 3 shows two colorings of the same graph, which is decomposed in three groups $V_1$, $V_2$ and $V_3$ containing nodes with attributes $a$, $b$ and $c$ respectively. Since $V_1$ contains only two nodes, a coloring of the graph will attribute two colors ($(1, 0)$ and $(0, 1)$, depicted as *blue* and *red*) to these nodes. More precisely, the set of colorings $\mathcal{C}(v, A)$ of a graph $G = (v, A)$ are defined as

$$\mathcal{C}(v, A) = \left\{ (c_1, ..., c_n) \ : \ \forall k \in [\![1, K]\!], (c_i)_{i \in V_k} \text{ is a permutation of } C_{|V_k|} \right\}. \tag{4}$$

### 5.2 THE CLIP ALGORITHM

In the CLIP algorithm, we add a coloring scheme to an MPNN in order to distinguish identical node attributes. This is achieved by modifying the initialization and readout phases of MPNNs as follows.

1. **Colored initialization:** We first select a set $\mathcal{C}_k \subseteq \mathcal{C}(v, A)$ of $k$ distinct colorings uniformly at random (see Eq. (4)). Then, for each coloring $c \in \mathcal{C}_k$, node representations are initialized with their node attributes concatenated with their color: $x_{i,0}^c = (v_i, c_i)$.

2. **Aggregation and combination:** This step is performed for all colorings $c \in \mathcal{C}_k$ using a universal set representation as the aggregation function: $x_{i,t+1}^c = \psi^{(t)}\big(x_{i,t}^c, \sum_{j \in \mathcal{N}_i} \varphi^{(t)}(x_{j,t}^c)\big)$,

where $\psi$ and $\varphi$ are MLPs with continuous non-polynomial activation functions and $\psi(x, y)$ denotes the result of $\psi$ applied to the concatenation of $x$ and $y$. The aggregation scheme we propose is closely related to DeepSet (Zaheer et al., 2017), and a direct application of Corollary 1 proves the universality of our architecture. More details, as well as the proof of universality, are available in Appendix C.

3. **Colored readout:** This step performs a maximum over all possible colorings in order to obtain a final *coloring-independent* graph representation. In order to keep the stability by concatenation, the maximum is taken coefficient-wise

$$x_G = \psi \left( \max_{c \in \mathcal{C}_k} \sum_{i=1}^{n} x_{i,T}^c \right) , \tag{5}$$

where $\psi$ is an MLP with continuous non polynomial activation functions.

We treat $k$ as a hyper-parameter of the algorithm and call $k$-CLIP (resp. $\infty$-CLIP) the algorithm using $k$ colorings (resp. all colorings, i.e. $k = |\mathcal{C}(v, A)|$). Note that, while our focus is graphs with node attributes, the approach used for CLIP is easily extendable to similar data structures such as directed or weighted graphs with node attributes, graphs with node labels, graphs with edge attributes or graphs with additional attributes at the graph level.

## 5.3 UNIVERSAL REPRESENTATION THEOREM

As the colorings are chosen at random, the CLIP representation is itself random as soon as $k < |\mathcal{C}(v, A)|$, and the number of colorings $k$ will impact the variance of the representation. However, $\infty$-CLIP is deterministic and permutation invariant, as MPNNs are permutation invariant. The separability is less trivial and is ensured by the coloring scheme.

**Theorem 3.** *The $\infty$-CLIP algorithm with one local iteration ($T = 1$) is a universal representation of the space $\mathbf{Graph}_m$ of graphs with node attributes.*

The proof of Theorem 3 relies on showing that $\infty$-CLIP is separable and applying Corollary 1. This is achieved by fixing a coloring on one graph and identifying all nodes and edges of the second graph using the fact that all pairs $(v_i, c_i)$ are dissimilar (see Appendix D). Similarly to the case of MLPs, only one local iteration is necessary to ensure universality of the representation. This rather counter-intuitive result is due to the fact that all nodes can be identified by their color, and the readout function can aggregate all the structural information in a complex and non-trivial way. However, as for MLPs, one may expect poor generalization capabilities for CLIP with only one local iteration, and deeper networks may allow for more complex representations and better generalization. This point is addressed in the experiments of Section 6. Moreover, $\infty$-CLIP may be slow in practice due to a large number of colorings, and reducing $k$ will speed-up the computation. Fortunately, while $k$-CLIP is random, a similar universality theorem still holds even for $k = 1$.

**Theorem 4.** *The 1-CLIP algorithm with one local iteration ($T = 1$) is a random representation whose expectation is a universal representation of the space $\mathbf{Graph}_m$ of graphs with node attributes.*

The proof of Theorem 4 relies on using $\infty$-CLIP on the augmented node attributes $v_i' = (v_i, c_i)$. As all node attributes are, by design, different, the max over all colorings in Eq. (5) disappears and, for any coloring, 1-CLIP returns an $\varepsilon$-approximation of the target function (see Appendix D).

**Remark 1.** Note that the variance of the representation may be reduced by averaging over multiple samples. Moreover, the proof of Theorem 4 shows that the variance can be reduced to an arbitrary precision given enough training epochs, although this may lead to very large training times in practice.

## 5.4 COMPUTATIONAL COMPLEXITY

As the local iterative steps are performed $T$ times on each node and the complexity of the aggregation depends on the number of neighbors of the considered node, the complexity is proportional to the number of edges of the graph $E$ and the number of steps $T$. Moreover, CLIP performs this iterative aggregation for each coloring, and its complexity is also proportional to the number of chosen colorings $k = |\mathcal{C}_k|$. Hence the complexity of the algorithm is in $O(kET)$.

Note that the number of all possible colorings for a given graph depends exponentially in the size of the groups $V_1, ..., V_K$,

$$|\mathcal{C}(v, A)| = \prod_{k=1}^{K} |V_k|!, \tag{6}$$

and thus $\infty$-CLIP is practical only when most node attributes are dissimilar. This worst case exponential dependency in the number of nodes can hardly be avoided for universal representations. Indeed, a universal graph representation should also be able to solve the graph isomorphism problem. Despite the existence of polynomial time algorithms for a broad class of graphs (Luks, 1982; Bodlaender, 1990), graph isomorphism is still quasi-polynomial in general (Babai, 2016). As a result, creating a universal graph representation with polynomial complexity for all possible graphs and functions to approximate is highly unlikely, as it would also induce a graph isomorphism test of polynomial complexity and thus solve a very hard and long standing open problem of theoretical computer science.

# 6 EXPERIMENTS

In this section we show empirically the practical efficiency of CLIP and its relaxation. We run two sets of experiments to compare CLIP w.r.t. state-of-the-art methods in supervised learning settings: i) on 5 real-world graph classification datasets and ii) on 4 synthetic datasets to distinguish structural graph properties and isomorphism. Both experiments follow the same experimental protocol as described in Xu et al. (2019): 10-fold cross validation with grid search hyper-parameter optimization. More details on the experimental setup are provided in Appendix E.

## 6.1 CLASSICAL BENCHMARK DATASETS

We performed experiments on five benchmark datasets extracted from standard social networks (IMDBb and IMDBm) and bio-informatics databases (MUTAG, PROTEINS and PTC). All dataset characteristics (e.g. size, classes), as well as the experimental setup, are available in Appendix E. Following standard practices for graph classification on these datasets, we use one-hot encodings of node degrees as node attributes for IMDBb and IMDBm (Xu et al., 2019), and perform single-label multi-class classification on all datasets. We compared CLIP with six state-of-the-art baseline algorithms: 1) **WL:** Weisfeiler-Lehman subtree kernel (Shervashidze et al., 2011), 2) **AWL:** Anonymous Walk Embeddings (Ivanov and Burnaev, 2018), 3) **DCNN:** Diffusion-convolutional neural networks (Atwood and Towsley, 2016), 4) **PS:** PATCHY-SAN (Niepert et al., 2016), 5) **DGCNN:** Deep Graph CNN (Zhang et al., 2018) and 6) **GIN:** Graph Isomorphism Network (Xu et al., 2019). WL and AWL are representative of unsupervised methods coupled with an SVM classifier, while DCNN, PS, DGCNN and GIN are four deep learning architectures. As the same experimental protocol as that of Xu et al. (2019) was used, we present their reported results on Table 1.

Table 1: Classification accuracies of the compared methods on benchmark datasets. The best performer w.r.t. the mean is highlighted with an asterisk. We perform an unpaired t-test with asymptotic significance of 0.1 w.r.t. the best performer and highlight with boldface the ones for which the difference is not statistically significant. 0-CLIP is the CLIP architecture without any colorings.

| Dataset | PTC | IMDBb | IMDBm | PROTEINS | MUTAG |
|---------|-----|-------|-------|----------|-------|
| **WL** | 59.9±4.3 | **73.8±3.9** | **50.9±3.8** | **75.0±3.1** | 90.4±5.7 |
| **DCNN** | 56.6 | 49.1 | 33.5 | 61.3 | 67.0 |
| **PS** | 60.0±4.8 | 71.0±2.2 | 45.2±2.8 | **75.9±2.8** | **92.6±4.2** |
| **DGCNN** | 58.6 | 70.0 | 47.8 | **75.5** | 85.8 |
| **AWL** | - | **74.5±5.9** | **51.5±3.6** | - | 87.9±9.8 |
| **GIN** | **64.6±7.0** | **75.1±5.1** | **52.3±2.8** | **76.2±2.8** | 89.4±5.6 |
| **0-CLIP** | **65.9±4.0** | **75.4±2.0** | **52.5±2.6***| **77.0±3.2** | 90.0±5.1 |
| **CLIP** | **67.9±7.1***| **76.0±2.7***| **52.5±3.0***| **77.1±4.4***| **93.9±4.0***|

As Table 1 shows, CLIP can achieve state-of-the-art performance on the five benchmark datasets. Moreover, CLIP is consistent across all datasets, while all other competitors have at least one weak

performance. This is a good indicator of the robustness of the method to multiple classification tasks and dataset types. Finally, the addition of colors does not improve the accuracy for these graph classification tasks, except on the MUTAG dataset. This may come from the small dataset sizes (leading to high variances) or an inherent difficulty of these classification tasks, and contrasts with the clear improvements of the method for property testing (see Section 6.2). More details on the performance of CLIP w.r.t. the number of colors $k$ are available in Appendix E.

**Remark 2.** In three out of five datasets, none of the recent state-of-the-art algorithms have statistically significantly better results than older methods (e.g. WL). We argue that, considering the high variances of all classification algorithms on classical graph datasets, graph property testing may be better suited to measure the expressiveness of graph representation learning algorithms in practice.

## 6.2 GRAPH PROPERTY TESTING

We now investigate the ability of CLIP to identify structural graph properties, a task which was previously used to evaluate the expressivity of graph kernels and on which the Weisfeiler-Lehman subtree kernel has been shown to fail for bounded-degree graphs (Kriege et al., 2018). The performance of our algorithm is evaluated for the binary classification of four different structural properties: 1) connectivity, 2) bipartiteness, 3) triangle-freeness, 4) circular skip links (Murphy et al., 2019) (see Appendix E for precise definitions of these properties) against three competitors: a) GIN, arguably the most efficient MPNN variant yet published (Xu et al., 2019), b) Ring-GNN, a permutation invariant network that uses the ring of matrix addition and multiplication (Chen et al., 2019), c) RP-GIN, the Graph Isomorphism Network combined with Relational Pooling, as described by Murphy et al. (2019), which is able to distinguish certain cases of non-isomorphic regular graphs. We provide all experimental details in Appendix E.

Table 2: Classification accuracies of the synthetic datasets. $k$-RP-GIN refers to a relational pooling averaged over $k$ random permutations. We report Ring-GNN results from Chen et al. (2019).

| Property | Connectivity mean $\pm$ std | Bipartiteness mean $\pm$ std | Triangle-freeness mean $\pm$ std | Circular skip links mean $\pm$ std | max | min |
|---|---|---|---|---|---|---|
| **GIN** | $55.2 \pm 4.4$ | $53.1 \pm 4.7$ | $50.7 \pm 6.1$ | $10.0 \pm 0.0$ | 10.0 | 10.0 |
| **Ring-GNN** | - | - | - | $(?) \pm 15.7$ | 80.0 | 10.0 |
| **1-RP-GIN** | $66.1 \pm 5.2$ | $66.0 \pm 5.1$ | $63.0 \pm 3.6$ | $20.0 \pm 7.0$ | 28.6 | 10.0 |
| **16-RP-GIN** | $83.3 \pm 7.9$ | $64.9 \pm 4.1$ | $65.7 \pm 3.3$ | $37.6 \pm 12.9$ | 53.3 | 10.0 |
| **0-CLIP** | $56.5 \pm 4.0$ | $55.4 \pm 5.7$ | $59.6 \pm 3.8$ | $10.0 \pm 0.0$ | 10.0 | 10.0 |
| **1-CLIP** | $73.3 \pm 2.2$ | $63.3 \pm 1.9$ | $63.5 \pm 7.3$ | $61.9 \pm 11.9$ | 80.7 | 36.7 |
| **16-CLIP** | $\mathbf{99.7 \pm 0.5}$ | $\mathbf{99.2 \pm 0.9}$ | $\mathbf{94.2 \pm 3.4}$ | $\mathbf{90.8 \pm 6.8}$ | 98.7 | 76.0 |

Table 2 shows that CLIP is able to capture the structural information of connectivity, bipartiteness, triangle-freeness and circular skip links, while MPNN variants fail to identify these graph properties. Furthermore, we observe that CLIP outperforms RP-GIN, that was shown to provide very expressive representations for regular graphs (Murphy et al., 2019), even with a high number of permutations (the equivalent of colors in their method is set to $k = 16$). Moreover, both for $k$-RP-GIN and $k$-CLIP, the increase of permutations and colorings respectively lead to higher accuracies. In particular, CLIP can capture almost perfectly the different graph properties with as little as $k = 16$ colorings.

## 7 CONCLUSION

In this paper, we showed that a simple coloring scheme can improve the expressive power of MPNNs. Using such a coloring scheme, we extended MPNNs to create CLIP, the first universal graph representation. Universality was proven using the novel concept of separable neural networks, and our experiments showed that CLIP is state-of-the-art on both graph classification datasets and property testing tasks. The coloring scheme is especially well suited to hard classification tasks that require complex structural information to learn. The framework is general and simple enough to extend to other data structures such as directed, weighted or labeled graphs. Future work includes more detailed and quantitative approximation results depending on the parameters of the architecture such as the number of colors $k$, or number of hops of the iterative neighborhood aggregation.

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

## A    PROOFS OF THE UNIVERSALITY OF SEPARABLE NEURAL NETWORKS

*Proof of Theorem 2.* The proof relies on the Stone-Weierstrass theorem we recall below. We refer to (Rudin, 1987, Theorem 7.32) for a detailed proof of the following classical theorem.

**Theorem 5** (Stone-Weierstrass). *Let $\mathcal{A}$ be an algebra of real functions on a compact Hausdorff set $K$. If $\mathcal{A}$ separates points of $K$ and contains a non-zero constant function, then $\mathcal{A}$ is uniformly dense in $\mathcal{C}(K, \mathbb{R})$.*

We verify that under the assumptions of Theorem 2 the Stone-Weierstrass theorem applies. In this setting, we first prove the theorem for $m = 1$ and use induction for the general case.

Let $K \subset \mathcal{X}$ be a compact subset of $\mathcal{X}$. We will denote

$$\mathcal{A}_0 = \{\psi \circ f \ : \ \exists d \geq 1 \text{ s.t. } \psi \in \mathcal{C}(\mathbb{R}^d, \mathbb{R}), f \in \mathfrak{F}\},$$

and will proceed in two steps: we first show that $\mathcal{A}_0$ is uniformly dense in $\mathcal{C}(K, \mathbb{R})$, then that $\mathcal{A}$ is dense in $\mathcal{A}_0$, hence proving Theorem 2.

**Lemma 1.** *$\mathcal{A}_0$ is a subalgebra of $\mathcal{C}(K, \mathbb{R})$.*

*Proof.* The subset $\mathcal{A}_0$ contains zero and all constants. Let $f, g \in \mathcal{A}_0$ so that

$$f(x) = \psi_f \circ \varphi_f(x), \qquad g(x) = \psi_g \circ \varphi_g(x),$$

with $\psi_f : \mathbb{R}^{d_f} \to \mathbb{R}$ and $\psi_g : \mathbb{R}^{d_g} \to \mathbb{R}$. Consider $\psi : \mathbb{R}^{d_f + d_g} \to \mathbb{R}$ such that $\psi(a, b) = \psi_f(a) + \psi_g(b)$. We define $\varphi(x) = (\varphi_f(x), \varphi_g(x)) \in \mathbb{R}^{d_f + d_g}$ and by assumption $\varphi \in \mathfrak{F}$. We have

$$\begin{aligned}(f + g)(x) &= \psi(\varphi_f(x), \varphi_g(x)) \\ &= \psi \circ \varphi(x)\end{aligned}$$

so that $f + g \in \mathcal{A}_0$ and we conclude that $\mathcal{A}_0$ is a vectorial subspace of $\mathcal{C}(K, \mathbb{R})$. We proceed similarly for the product in order to finish the proof of the lemma. $\square$

Because $\mathfrak{F}_1$ separates the points of $\mathcal{X}$ by assumption, $\mathcal{A}_0$ also separates the points of $\mathcal{X}$. Indeed, let $x \neq y$ two distinct points of $X$ so that $\exists f \in \mathfrak{F}$ such that $f(x) \neq f(y)$. There exists $g \in \mathcal{C}(\mathbb{R}^d, \mathbb{R})$ such that $g(f(x)) \neq g(f(y))$. From Theorem 5 we deduce that $\mathcal{A}_0$ is uniformly dense in $\mathcal{C}(K, \mathbb{R})$ for all compact subsets $K \subset \mathcal{X}$.

Finally we state that:

**Lemma 2.** *For any compact subset $K \subset \mathcal{X}$, $\mathcal{A}$ is uniformly dense in $\mathcal{A}_0$.*

*Proof.* Let $\epsilon > 0$ and $h = \psi_0 \circ f \in \mathcal{A}_0$ with $f \in \mathfrak{F}$ and $\psi_0 \in \mathcal{C}(\mathbb{R}^d, \mathbb{R})$. Thanks to the continuity of $f$, the image $\tilde{K} = f(K)$ is a compact of $\mathbb{R}^d$. By Theorem 1 there exists an MLP $\psi$ such that $\|\psi - \psi_0\|_{\tilde{K}, \infty} \leq \epsilon$. We have $\psi \circ f \in \mathcal{A}$ and $\|\psi_0 \circ f - \psi \circ f\|_{K, \infty} \leq \epsilon$ which concludes the proof. $\square$

This last lemma completes the proof in the case $m = 1$. For $m \geq 2$ consider $\mathcal{A}_0 = \{\psi \circ f \ : \ \exists d \geq 1 \text{ s.t. } \psi \in \mathcal{C}(\mathbb{R}^d, \mathbb{R}^m), f \in \mathfrak{F}\}$ and proceed in a similar manner than Lemma 2 by decomposing $\psi \in \mathcal{C}(\mathbb{R}^d, \mathbb{R}^m)$ as

$$\psi(x) = \begin{pmatrix} \psi_1(x) \\ \psi_2(x) \\ \vdots \\ \psi_m(x) \end{pmatrix},$$

and applying Lemma 1 for each coefficient function $\psi_i \in \mathcal{C}(\mathbb{R}^d, \mathbb{R})$. $\square$

*Proof of Proposition 1.* Assume that there exists $x, y \in \mathcal{X}$ s.t. $\forall f \in \mathfrak{F}_1, f(x) = f(y)$. Then $K = \{x, y\}$ is a compact subset of $\mathcal{X}$ and let $\phi \in \mathcal{C}(K, \mathbb{R})$ be such that $\phi(x) = 1$ and $\phi(y) = 0$. Thus, for all $f \in \mathfrak{F}_1, \max_{z \in \{x, y\}} \|\phi(z) - f(z)\| \geq 1/2$ which contradicts universality (see Definition 1). $\square$

## B  GROUP ACTION ON HAUSDORFF SPACES

In what follows, $\mathcal{X}$ is always a topological set and $G$ a group of transformations acting on $\mathcal{X}$. The orbits of $\mathcal{X}$ under the action of $G$ are the sets $Gx = \{g \cdot x : g \in G\}$. Moreover, we denote as $\mathcal{X}/G$ the quotient space of orbits, also defined by the equivalence relation: $x \sim y \iff \exists g \in G$ s.t. $x = g \cdot y$. As stated in Section 5, graphs with node attributes can be defined using invariance by permutation of the labels. We prove here that the resulting spaces are Hausdorff.

**Definition 4** (Group invariance). Let $G$ a group, a function $f : \mathcal{X} \to \mathcal{Y}$ is *G-invariant* if

$$\forall x \in \mathcal{X}, \forall g \in G, f(x) = f(g \cdot x). \tag{7}$$

**Lemma 3** ((Bourbaki, 1998, I, §8.3)). *Let $\mathcal{X}$ be a Hausdorff space and $\mathcal{R}$ an equivalence relation of $\mathcal{X}$. Then $\mathcal{X}/\mathcal{R}$ is Hausdorff if and only if any two distinct equivalence classes in $\mathcal{X}$ are contained in disjoints saturated open subsets of $\mathcal{X}$.*

Thanks to this lemma we prove the following proposition.

**Proposition 2.** *Let $G$ a finite group acting on an Hausdorff space $\mathcal{X}$, then the orbit space $\mathcal{X}/G$ is Hausdorff.*

*Proof.* Let $Gx$ and $Gy$ two distinct classes with disjoint open neighbourhood $U$ and $V$. By finiteness of $G$, the application $\pi : \mathcal{X} \to \mathcal{X}/G$ is open, hence the saturated sets $\tilde{U} = \pi^{-1}[\pi(U)]$ and $\tilde{V} = \pi^{-1}[\pi(V)]$ are open. Suppose that there exists $z \in \tilde{U} \cap \tilde{V}$, then $\pi(z) \in \pi(U) \cap \pi(V)$ and we finally get that $Gz \subset U \cap V = \emptyset$. Therefore $\tilde{U} \cap \tilde{V}$ is empty and $\mathcal{X}/G$ is Hausdorff by Lemma 3. □

Proposition 2 directly implies that the spaces $\mathbf{Graph}_m$ and $\mathbf{Neighborhood}_m$ are Hausdorff.

## C  UNIVERSALITY OF THE NODE AGGREGATION SCHEME

We now provide more details on the aggregation and combination scheme of CLIP, and show that a simple application of Corollary 1 is sufficient to prove its universality for node neighborhoods. Each local aggregation step takes as input a couple $(x_i, \{x_j\}_{j \in \mathcal{N}_i})$ where $x_i \in \mathbb{R}^m$ is the representation of node $i$, and $\{x_j\}_{j \in \mathcal{N}_i}$ is the set of vector representations of the neighbors of node $i$. In the following, we show how to use Corollary 1 to design universal representations for node neighborhoods.

**Definition 5.** The set of node neighborhoods for $m$-dimensional node attributes is defined as

$$\mathbf{Neighborhood}_m = \mathbb{R}^m \times \bigcup_{n \leq n_{\max}} \left( \mathbb{R}^{n \times m} / \mathcal{P}_n \right), \tag{8}$$

where the set of permutation matrices $\mathcal{P}_n$ is acting on $\mathbb{R}^{n \times m}$ by $P \cdot v = Pv$.

The main difficulty to design universal neighborhood representations is that the node neighborhoods of Definition 5 are permutation invariant w.r.t. neighboring node attributes, and hence require permutation invariant representations. The graph neural network literature already contains several deep learning architectures for permutation invariant sets (Guttenberg et al., 2016; Qi et al., 2017; Zaheer et al., 2017; Xu et al., 2019), among which PointNet and DeepSet have the notable advantage of being provably universal for sets. Following Corollary 1, we compose a separable permutation invariant network with an MLP that will aggregate both information from the node itself and its neighborhood. While our final architecture is similar to Deepset (Zaheer et al., 2017), this section emphasizes that the general universality theorems of Section 3 are easily applicable in many settings including permutation invariant networks. The permutation invariant set representation used for the aggregation step of CLIP is as follows:

$$\text{NODEAGGREGATION}(x, S) = \psi \left( x, \sum_{y \in S} \varphi(y) \right), \tag{9}$$

where $\psi$ and $\varphi$ are MLPs with continuous non-polynomial activation functions and $\psi(x, y)$ denotes the result of the MLP $\psi$ applied to the concatenation of $x$ and $y$.

**Theorem 6.** *The set representation described in Eq. (9) is a universal representation of* $\mathbf{Neighborhood}_m$.

*Proof.* By construction, NODEAGGREGATION is a continuous and concatenable representation. Moreover, its final stage is an MLP, and we thus only have to prove separability in order to use Corollary 1 and prove universality. Let $(x^1, S^1), (x^2, S^2) \in \mathbf{Neighborhood}_m$ and suppose that $(x^1, S^1) \neq (x^2, S^2)$. First, if $x^1 \neq x^2$, the final MLP $\psi$ can separate $x^1$ and $x^2$. Otherwise, $S^1 \neq S^2$, and let us assume that $S^1 \setminus S^2 \neq \emptyset$ (otherwise $S^2 \setminus S^1 \neq \emptyset$ and the argument is identical). Since MLPs are universal representations of $\mathbb{R}^m$, there exists an MLP $\varphi$ such that, $\forall s \in S^1 \cup S^2$,

$$\varphi(s) \geq 1 \ \text{if} \ s \in S^1 \setminus S^2 \,,$$
$$|\varphi(s)| \leq \varepsilon \ \text{otherwise} \,,$$

Taking $\psi(x, y) = y$ and $\varepsilon = 1/3 \max\{|S^1|, |S^2|\}$, we have

$$\text{NODEAGGREGATION}(x^1, S^1) \geq 2/3 \,,$$
$$\text{NODEAGGREGATION}(x^2, S^2) \leq 1/3 \,,$$

which proves separability and, using Corollary 1, the universality of the representation. $\qquad\square$

## D    PROOF OF THE UNIVERSALITY OF CLIP

*Proof of Theorem 3.* First of all, as the activation functions of the MLPs are continuous, CLIP is made of continuous and concatenable functions, and is thus also continuous and concatenable. Second, as the node aggregation step (denoted NODEAGGREGATION below) is a universal set representation (see Appendix C), it is capable of approximating any continuous function. We will thus first replace this function by a continuous function $\phi$, and then show that the result still holds for NODEAGGREGATION$^{(1)}$ by a simple density argument. Let $G^1 = (v^1, A^1)$ and $G^2 = (v^2, A^2)$ be two distinct graphs of respective sizes $n_1$ and $n_2$ (up to a permutation). If $n^1 \neq n^2$, then $\psi(x) = x$ and $\phi(x) = 1$ returns the number of nodes, and hence $x_{G^1} = n^1 \neq n^2 = x_{G^2}$. Otherwise, let $V = \{v_i^k\}_{i \in [\![1, n^1]\!], k \in \{1,2\}}$ be the set of node attributes of $G^1$ and $G^2$, $c^1$ be a coloring of $G^1$, $\psi(x) = x$ and $\phi$ be a continuous function such that, $\forall x \in V$ and $S \subset V$,

$$\phi(x, S) = \sum_{i=1}^{n^1} \mathbb{1}\{x = (v_i^1, c_i^1)\} \prod_{j \neq i} \mathbb{1}\left\{A_{ij}^1 = \mathbb{1}\{(v_j^1, c_j^1) \in S\}\right\} . \tag{10}$$

The existence of $\phi \in \mathcal{C}(\mathbb{R}^m, \mathbb{R})$ is assured by Urysohn's lemma (see e.g. (Rudin, 1987, lemma 2.12)). Then, $x_G$ counts the number of matching neighborhoods for the best coloring, and we have $x_{G^1} = n^1$ and $x_{G^2} \leq n^1 - 1$. Finally, taking $\varepsilon < 1/2n^1$ in the definition of universal representation leads to the desired result, as then, using an $\varepsilon$-approximation of $\phi$ as NODEAGGREGATION$^{(1)}$, we have $x_{G^1} > n^1 - 1/2 > x_{G^2}$. $\qquad\square$

*Proof of Theorem 4.* Consider a continuous function $\psi : \mathbf{Graph}_m \to \mathbb{R}^d$ and a compact $K' \subset \mathbf{Graph}_m$. Let extend $K'$ with $K = K' \times [0, 1]^{n_{\max}}$ and we define $\phi : \mathbf{Graph}_{m+n_{\max}} \to \mathbb{R}^d$ with $\phi((v, c), A) = \psi(v, A)$ for all $c \in \mathcal{C}(v, A)$. Since $\infty$-CLIP is universal there exists $f \in \infty$-CLIP such that, for all $((v, c), A) \in K$,

$$\|\phi((v, c), A) - f((v, c), A)\| \leq \varepsilon \,, \tag{11}$$

hence

$$\|\psi(v, A) - f((v, c), A)\| \leq \varepsilon \,. \tag{12}$$

Moreover, observe that for any coloring $c \in \mathcal{C}(v, A)$, $\infty$-CLIP and 1-CLIP applied to $((v, c), A)$ returns the same result, as all node attributes are dissimilar (by definition of the colorings) and $\mathcal{C}((v, c), A) = \emptyset$. Finally, 1-CLIP applied to $(v, A)$ is equivalent to applying 1-CLIP to $((v, C), A)$ where $C$ is a random coloring in $\mathcal{C}(v, A)$, and Eq. (12) thus implies that any random sample of 1-CLIP is within an $\varepsilon$ error of the target function $\psi$. As a result, its expectation is also within an $\varepsilon$ error of the target function $\psi$, which proves the universality of the expectation of 1-CLIP. $\qquad\square$

# E  EXPERIMENTAL DETAILS

## E.1  REAL-WORLD DATASETS

Table 3 summarizes the characteristics of all benchmark graph classification datasets used in Section 6.1. We now provide complementary information on these datasets.

**Social Network Datasets (IMDBb, IMDBm):** These datasets refer to collaborations between actors/actresses, where each graph is an ego-graph of every actor and the edges occur when the connected nodes/actors are playing in the same movie. The task is to classify the genre of the movie that the graph derives from. IMDBb is a single-class classification dataset, while IMDBm is multi-class. For both social network datasets, we used one-hot encodings of node degrees as node attribute vectors.

**Bio-informatics Datasets (MUTAG, PROTEINS, PTC):** MUTAG consists of mutagenic aromatic and heteroaromatic nitrocompounds with 7 discrete labels. PROTEINS consists of nodes, which correspond to secondary structureelements and the edges occur when the connected nodes are neighbors in the amino-acidsequence or in 3D space. It has 3 discrete labels. PTC consists of chemical compounds that reports the carcinogenicity for male and female rats and it has 19 discrete labels. For all bio-informatics datasets we used the node labels as node attribute vectors.

**Experimentation protocol:** We follow the same experimental protocol as described in Xu et al. (2019), and thus report the results provided in this paper corresponding to the accuracy of our six baselines in Table 1. We optimized the CLIP hyperparameters by grid search according to 10-fold cross-validated accuracy means. We use 2-layer MLPs, an initial learning rate of $0.001$ and decreased the learning rate by 0.5 every 50 epochs for all possible settings. For all datasets the hyperparameters we tested are: the number of hidden units within $\{32, 64\}$, the number of colorings $c \in \{1, 2, 4, 8\}$, the number of MPNN layers within $\{1, 3, 5\}$, the batch size within $\{32, 64\}$, and the number of epochs, that means, we select a single epoch with the best cross-validation accuracy averaged over the 10 folds. Note that standard deviations are fairly high for all models due to the small size of these classic datasets.

Table 3: Characteristics of the benchmark graph classification datasets used in Section 6.1.

| Dataset | PTC | IMDBb | IMDBm | PROTEINS | MUTAG |
|---|---|---|---|---|---|
| **# graphs** | 344 | 1000 | 1500 | 1113 | 188 |
| **# classes** | 2 | 2 | 3 | 2 | 2 |
| **Avg # nodes** | 14.29 | 19.77 | 13.00 | 39.06 | 17.93 |
| **Avg degree** | 2.05 | 9.76 | 10.14 | 3.72 | 2.21 |

### E.1.1  CLIP PERFORMANCES W.R.T. THE NUMBER OF COLORINGS $k$

Table 4 summarizes the performances of CLIP while increasing the number of colorings $k$. Overall we can see a small increase in performances and a reduction of the variances when $k$ is increasing. Nevertheless we should not jump to any conclusions since none of the models are statistically significantly better than the others.

Table 4: Ablation study: classification accuracies of $k$-CLIP on benchmark datasets w.r.t $k$.

| Dataset | PTC | IMDBb | IMDBm | PROTEINS | MUTAG |
|---|---|---|---|---|---|
| **0-CLIP** | 65.9±4.0 | 75.4±2.0 | 52.5±2.6 | 77.0±3.2 | 90.0±5.1 |
| **1-CLIP** | 65.3±12.8 | 75.2±3.9 | 52.2±4.0 | 75.1±4.5 | 91.1±7.0 |
| **4-CLIP** | 65.9±5.7 | 75.8±5.0 | 51.8±2.9 | 77.1±4.4 | 92.2±7.0 |
| **8-CLIP** | 67.9±7.1 | 75.7±3.8 | 52.5±3.0 | 76.8±4.8 | 93.9±4.1 |
| **16-CLIP** | 66.5±5.4 | 76.0±2.7 | 52.5±4.5 | 76.6±2.8 | 91.7±6.0 |

We note that on the IMDBb and PROTEINS datasets the difference between using or not a coloring scheme does not have a big impact on the performances. However, adding colors increases the

performances of the algorithm on three out of five real world datasets. The property testing section (Section 6.2) shows empirically that the color scheme improves the expressiveness of CLIP.

## E.2 GRAPH PROPERTY TESTING

In Section 6.2 we evaluate the expressive power of CLIP on benchmark synthetic datasets. Our goal is to show that CLIP is able to distinguish basic graph properties, where classical MPNN cannot. We considered a binary classification task and we constructed *balanced* synthetic datasets[2] for each of the examined graph properties. The 20-node graphs are generated using Erdös-Rényi model (Erdös and Rényi, 1959) (and its bipartite version for the bipartiteness) with different probabilities $p$ for edge creation. All nodes share the same (scalar) attribute. We thus have uninformative feature vectors.

In particular, we generated datasets for different classical tasks Kriege et al. (2018): 1) connectivity, 2) bipartiteness, 3) triangle-freeness, and 4) circular skip links (Murphy et al., 2019). In the following, we present the generating protocol of the synthetic datasets and the experimentation setup we used for the experiments.

**Synthetic datasets:**
In every case of synthetic dataset we follow the same pattern: we generate a set of random graphs using Erdös-Rényi model, which contain a specific graph property and belong to the same class and by proper edge addition we remove this property, thus creating the second class of graphs. By this way, we assure that we do not change different structural characteristics other than the examined graph property.

- **Connectivity dataset:** this dataset consists of 1000 (20-node) graphs with 500 positive samples and 500 negative ones. The positive samples correspond to disconnected graphs with two 10-node connected components selected among randomly generated graphs with an Erdös-Rényi model probability of $p = 0.5$. We constructed negative samples by adding to positive samples a random edge between the two connected components.

- **Bipartiteness dataset:** this dataset consists of 1000 (20-node) graphs with 500 positive samples and 500 negative ones. The positive samples correspond to bipartite graphs generated with an Erdös-Rényi (bipartite) model probability of $p = 0.5$. For the negative samples (non-bipartite graphs) we chose the positive samples and for each of them we added an edge between randomly selected nodes from the same partition, in order to form odd cycles [3].

- **Triangle-freeness dataset:** this dataset consists of 1000 (20-node) graphs with 500 positive samples and 500 negative ones. The positive samples correspond to triangle-free graphs selected among randomly generated graphs with an Erdös-Rényi model probability of $p = 0.1$. We constructed negative samples by randomly adding new edges to positive samples until it creates at least one triangle.

- **Circular skip links:** this dataset consists of 150 graphs of 41 nodes as described in (Murphy et al., 2019; Chen et al., 2019). The Circular Skip Links graphs are undirected regular graphs with node degree 4. We denote a Circular skip link graph by $G_{n,k}$ an undirected graph of $n$ nodes, where $(i, j) \in E$ holds if and only if $|i - j| \equiv 1$ or $k( \mod n)$ This is a 10-class multiclass classification task whose objective is to classify each graph according to its isomorphism class.

**Experimentation protocol:** We evaluate the different configurations of CLIP and its competitors GIN and RP-GIN based on their hyper-parameters. For the architecture implementation of the GIN, we followed the best performing architecture, presented in Xu et al. (2019). In particular, we used the summation as the aggregation operator, MLPs as the combination level for the node embedding generation and the sum operator for the readout function along with its refined version of concatenated graph representations across all iterations/layers of GIN, as described in Xu et al. (2019).
In all the tested configurations for CLIP and its competitors (GIN, RP-GIN) we fixed the number of layers of the MLPs and the learning rate: we chose 2-layer MLPs and we used the Adam optimizer with initial learning rate of $0.001$ along with a scheduler decaying the learning rate by $0.5$ every $50$ epochs. Concerning the other hyper-parameters, we optimized: the number of hidden units within $\{16, 32, 64\}$ (except for the CSL task where we only use 16 hidden units to be fair w.r.t. RP-GIN

---

[2]The datasets are available upon request.
[3]Having an odd cycle in a graph makes the graph non bipartite.

and Ring-GNN benchmarks), the number of MPNN layers within $\{1, 2, 3, 5\}$, the batch size within $\{32, 64\}$, and ran the model over $400$ epochs. Regarding the RP-GIN architecture (Murphy et al., 2019) we optimized the one-hot encoding dimension of the first update within $\{5, 10, 15, 20, 25, 30\}$ and the number of inference permutations within $\{1, 5, 16\}$. Regarding the CLIP algorithm, we optimized the number of colorings $c \in \{1, 2, 4, 8, 16\}$. We then performed a 10-fold cross validation with early stopping for the hyper-parameter optimization and we reported the best 10-fold cross-validated mean accuracy with its associated standard deviation.

