# OpenReview forum: "Coloring graph neural networks for node disambiguation"
_ICLR.cc/2020/Conference — Reject_

### Official Review · AnonReviewer2 · 2019-10-23
**Official Blind Review #2**

**Rating:** 1

**Review:**


This paper proposes a coloring scheme that can increase the expressive power of GCNs. Based on this coloring scheme, a colored local iterative procedure is built. Experimental studies are performed and demonstrate the effectiveness of the methods.

1. A major concern for this method is the permutation invariant in coloring scheme. In this work, nodes in a group is colored randomly. This means the graph will change with different coloring patterns. In section 5.3, inf-CLIP is claimed to be permutation invariant. However, this property can not be guaranteed for a normal k.

2. The experimental studies are weak. There should be some ablation studies to evaluate the effectiveness of the coloring scheme. In section 7.2, the ablation studies are performed on synthetic datasets. Why not use real data?

3. This paper exceeds 8 pages which means higher requirements are needed. The novelty of this paper is incremental and not technically sound.

Suggestions:

Figure out ways to ensure permutation invariant would be a great plus.

**Experience Assessment:**

I have published in this field for several years.

**Review Assessment: Checking Correctness Of Derivations And Theory:**

I assessed the sensibility of the derivations and theory.

**Review Assessment: Checking Correctness Of Experiments:**

I assessed the sensibility of the experiments.

**Review Assessment: Thoroughness In Paper Reading:**

I read the paper at least twice and used my best judgement in assessing the paper.

---

> ### Author Response · Authors · 2019-11-07
> **Review answer**
>
> Thank you very much for your review! We will update the paper in the next few days to address your concerns.
>
> 1. While k-CLIP is indeed not, strictly speaking, permutation invariant due to its randomness, note that its probability distribution is permutation invariant. Hence, k-CLIP can be seen as a permutation invariant representation with added (unbiased) noise. A discussed in Remark 1, the variance of k-CLIP may be reduced (and thus come closer to a deterministic representation) by averaging over multiple independent samples of the representation. Of course, this incurs an additional computation cost, and experimental evaluations suggest that the noise of k-CLIP remains sufficiently small in practice.
> The difficult question of finding deterministic representations that are both computationally tractable and provably universal is very interesting and left for future work. However, we discuss at the end of Section 5.4 the fact that such a universal and tractable graph representation may not exist, as it would also solve the graph isomorphism problem in polynomial time: a notoriously difficult problem in graph theory.
>
> 2. In the experiments on real graph classification datasets, the number of colors did not have a large impact on the accuracy (as for other competitors such as RP-GIN), and we thus decided not to report these results to gain a little extra space. However, it is true that this is an important information for readers, and we will add a short discussion to the experimental section as well as the results of 0-CLIP (without any coloring), 1-CLIP and 16-CLIP to the appendix. Moreover, we will also add another experiment used by a recent related paper [1] to assess the quality of universal graph representations.
>
> 3. The main novelty of this paper is to provide a theoretical analysis of universality for graph neural networks. The proposed algorithm in Section 5 is a direct application of the theory of separable neural networks (a novel concept defined in Section 3.3) that we develop in Section 3 and 4 as well as in the supplementary material. We give another use case of our theoretical framework in Section 6 for permutation invariant sets (this section will be removed). To the best of our knowledge, Theorem 2 with this level of generality is a novel result and does not appear in the GNN literature (as well as the following propositions and corollaries).
> Our experiments on graph classification and property testing benchmarks show that, unlike most deep learning SOTA algorithms, our method is able to reach state of the art results in these two relatively different tasks.
>
> 4. We apologies for the additional page, and decided to replace Section 6 by a remark in Section 5 in order to meet the eight page limit. This section was, as correctly noted by Reviewer #1, too long and mostly containing already known concepts.
>
> [1] Chen, Z., Villar, S., Chen, L. and Bruna, J., 2019. On the equivalence between graph isomorphism testing and function approximation with GNNs. arXiv preprint arXiv:1905.12560.

---

### Official Review · AnonReviewer3 · 2019-10-23
**Official Blind Review #3**

**Rating:** 6

**Review:**


The paper presents an interesting work, called Colored Local Iterative Procedure (CLIP), to improve the expressive power of Message Passing Neural Networks (MPNNs). Considering the expressive power from the concept of universal representations, the authors introduced the concept of separability and combine the separable representation with MLP to achieve the universal representation for graphs. They then developed a coloring scheme to improve the MPNN, and obtained superior performance on benchmark graph classification datasets as well as in the graph property testing experiments. In general, I like the paper, but I have the following concerns:

Although we can easily get the idea that universal representation is more expressive, however, I did feel a small conceptual gap between isomorphism test and universal representation. For example, in Section 4.2, when the authors talked about the fact that MPNN is not expressive to construct isomorphism tests for a k-regular graph, it is expected to have a more explicit explanation of how universal representations can solve this and how it is connected to isomorphism test. It seems that there is no such explanation in the paper.

I am not very clear about how 1-CLIP gets the randomness. To my understanding, 1-CLIP uses one color, so the identical node attributes still have the same node attributes after coloring, and it is essentially equivalent to just concatenating extra node features to an MPNN? It also does not change the expressive power of MPNN.

Intuitively k-CLIP should be better than p-CLIP if k>p, and it is also demonstrated in the graph property testing experiment. However, why do the authors use k as a hyperparameter to select the best results in classical benchmark datasets? Does it say sometimes the smaller k can also get a better result? Why not also just show the results of 1-CLIP and 16-CLIP?

It seems $k$ has different meanings in different places of the paper. For example, $k$ in $C_k$ is different the $k$ in Eq. (4). Maybe it is better to use a different variable to avoid confusion.


**Experience Assessment:**

I have published one or two papers in this area.

**Review Assessment: Checking Correctness Of Derivations And Theory:**

I did not assess the derivations or theory.

**Review Assessment: Checking Correctness Of Experiments:**

I assessed the sensibility of the experiments.

**Review Assessment: Thoroughness In Paper Reading:**

I read the paper at least twice and used my best judgement in assessing the paper.

---

> ### Author Response · Authors · 2019-11-07
> **Review answer**
>
> Thank you very much for your review! We will update the paper in the next few days to address your concerns.
>
> 1. The universal representation and isomorphism problem are indeed closely related (Reviewer #1 pointed out [1] which may be of interest for you). More precisely, as stated in Proposition 1, a universal representation of a graph G is able to separate points in the domain space (here a space of graphs). K-regular graphs are a simple example of a class of graphs that cannot be distinguished using classical MPNNs (there are many non-isomorphic k-regular graphs, see e.g. [2]). The property testing example from Section 7.2 highlights this point.
> Our paper shows in particular that inf-CLIP is able to separate, i.e. distinguish, all graphs up to a permutation (i.e. non-isomorphic) and so is our relaxed 1-CLIP algorithm in expectation.
>
> 2. The source of randomness of 1-CLIP comes from the fact that we add a single color to every node, the color being chosen randomly. It is thus similar to the addition of noise to every node attribute of the graph. Theorem 3 states that 1-CLIP is universal in expectation, although its variance may be large in practice. However, our experiments in Section 7 indicate that the variance remains relatively small, while being more expressive than classical MPNNs (see Section 7.2 and the property testing task).
>
> 3. In our experiments on real datasets, we think that the variance of the 10-fold cross validation accuracy is probably larger than the improvement due to an increase in colors. There was thus no visible improvement to using more colors, and we decided not to display these results in the paper to gain a little extra space. We will add the results of 0-CLIP (without any coloring), 1-CLIP and 16-CLIP to the appendix.
>
> 4. Thank you for pointing out these conflicting notations, we will take care of this in the updated version of the paper.
>
> [1] Chen, Z., Villar, S., Chen, L. and Bruna, J., 2019. On the equivalence between graph isomorphism testing and function approximation with GNNs. arXiv preprint arXiv:1905.12560.
> [2] https://oeis.org/A051031

---

### Official Review · AnonReviewer1 · 2019-10-24
**Official Blind Review #1**

**Rating:** 3

**Review:**

This paper proposes an extension of MPNN which leverages the random color augmentation to improve the representation power of MPNN. Authors also prove that two variants of the proposed method have universal representation power (one is exact and the other holds in expectation) from the separability perspective. Experiments on some small graph benchmark datasets and structural property tests are reported.

Overall, the paper seems to make a good contribution on advocating a new perspective of representation power of GNNs, i.e., separability, and proposes a variant to empirically improve representation power. However, I do have quite a few concerns listed as below which impedes my understanding and prevents me from giving a high score.

Pros:

1, The separability perspective of representation power seems novel.

2, The coloring based method is interesting and simple to implement.

3, The graph property test experiments are good testbeds to verify the representation power of various GNNs.

Cons & Questions:

1, The overall paper seems lack of focus in a sense that section 3 and 4 discuss too much on general universality whereas the main contribution, i.e., section 5 is not explained clearly.

2, If I understood correctly, the max operator in Eq. (5) only aggregates the “colored” representation within the group of nodes which shared the same attributes. How do you further get the representation of the whole graph? When k=1, the max operator in Eq. (5) becomes identity, wouldn’t 1-CLIP method be equivalent to augmenting random color as extra node features to GNNs?

3, The whole section 6 is just a very common GNN aggregation operator, I do not understand why authors claim it as “a novel universal neighborhood representation”. Also, the notation in Eq. (8) is not rigorous, what do you exactly mean by psi(x, y) as an MLP? Do you mean concatenating x and y as an input to MLP?

4, The experimental results on the benchmark datasets are less impressive as the mean performances are close to the WL-test results and the variances are considerably large. Moreover, why is the MPNN baseline missing, not mentioning other state-of-the-art GNNs? Same GNN baselines are missing in the structural property tests as well.

5, A closely relevant reference [1] is missing. The equivalence between universal approximation and graph isomorphism testing is studied in [1]. I think it is necessary to discuss the relationship. A comparison with [1] both theoretically and empirically would be make the paper more convincing.

6, Since k-CLIP with some k such that 1 < k < infinity achieves the best performance in the experiments, does k-CLIP still have universal representation theoretically?

7, Many notations are introduced without clear explanation. For example, what does lower-case c stand for? If it stands for the color per node, why does permutation appears in the definition of Eq. (4)? If I understood correctly, Eq. (4) is the set of all colorings which does not depend on permutation anyway. What does S refer to in Eq. (8)?

8, What are variants of CLIP reported in Table 1? Are they 1-CLIP? Also, the multiple bold numbers in Table 1 are quite confusing.

9, Wouldn't Eq. (6) indicate factorial growth rather than the claimed exponential one?

Typos: CDNN in table 1 should be DCNN

[1] Chen, Z., Villar, S., Chen, L. and Bruna, J., 2019. On the equivalence between graph isomorphism testing and function approximation with GNNs. arXiv preprint arXiv:1905.12560.

**Experience Assessment:**

I have published in this field for several years.

**Review Assessment: Checking Correctness Of Derivations And Theory:**

I assessed the sensibility of the derivations and theory.

**Review Assessment: Checking Correctness Of Experiments:**

I assessed the sensibility of the experiments.

**Review Assessment: Thoroughness In Paper Reading:**

I read the paper at least twice and used my best judgement in assessing the paper.

---

> ### Author Response · Authors · 2019-11-07
> **Review answer (1)**
>
> Thank you very much for your review! We will update the paper in the next few days to address your concerns.
>
> 1. In this work, we aim to give a complete and thorough study of universality in the context of GNNs. Hence the theoretical framework that allows a precise definition of CLIP is needed, also for pedagogical reasons. However, in the coming revised version we will give a better balanced version between the generalities and the clear description of our proposed algorithm.
>
> 2. There are 3 steps in order to compute the graph representation from the node representations. Firstly, for each color $c$ among the $k$ colorings of the graph we compute a color dependent graph representation (which correspond to the sums, given $c$, in Eq. (5)). Secondly, to be color independent, we compute a vector from the $k$ previous ones by taking the coefficient-wise maximum among them. Finally, we use an MLP $\psi$ that outputs the final graph representation $x_G$. We will make that clearer in the paper.
> Concerning your second remark, indeed, as you correctly understood when $k=1$, the max operator becomes an identity and we append a random coloring to the node attributes vector. That does not mean that we add a random color to every node attribute, but that we assign only once a different color randomly for the nodes with identical attributes.
>
> 3. We apologies for the strong claim of novelty in Section 6, and agree that most of the ideas of NeighborNet are already present in the literature. We thus decided to replace this section by a comment in Section 5 that shows that the universality of this known architecture for permutation invariant sets is a straightforward consequence of Corollary 1.
> Concerning your second comment, $\psi(x,y)$ indeed stands for the application of the function $\psi$ to the concatenation of $x$ and $y$.
>
> 4. We only kept one of the state of the art MPNN variants which is Graph Isomorphism Network (GIN) [2]. It has superior performance among other standard variants of MPNNs.
> Looking at the experiments, the large variances are not specific to our algorithm. However, our algorithm is the only one consistent across all datasets and, contrary to GIN, we are statistically better than WL on 2 out of 3 datasets (none for the state of the art MPNN can achieve this).
>
> 5. Thank you very much for pointing out this missing reference we were not aware of (it will be published at NeurIPS 2019). This work shares some similarities with ours and we are definitely going to cite and discuss it in our article.
> They introduce Ring-GNN wich is an equivariant neural network method based on [3]. While Ring-GNN has more expressive power than WL-1, one cannot make it straightforwardly a universal GNN.
> Note that CLIP also only uses 2-tensors (and inf-CLIP reaches universality).
>
> We will add the benchmark on Circular Skip Links to compare with our method as it is done in [4,5]. Looking at their reported experiments, early results show that CLIP significantly outperforms both RP-GIN and Ring-GNN. We will introduce this benchmark in the soon revised version of our paper.
>
> 6. k-CLIP has universal representation theoretically for k large enough. In section 5.3 we give a precise bound.
>
> 7. Thank you very much for pointing out these unclear notations.
> Indeed in 5.1 there may be some confusion as we first use 'c' in an example before using it more generally in Eq. (4). The chosen set of color $C = \{c_1, ..., c_n\}$ is ordered in all possible ways for the purpose of CLIP as stated in Eq. (4).
> In Eq. (8), $x$ is a vector and S a set of vectors. In the case of CLIP, x will be a feature vector of a node and S the set of feature vectors of its neighbors.
>
> [2] How powerful are graph neural networks, Xu et al., ICLR 2019, https://arxiv.org/pdf/1810.00826.pdf
> [3] Invariant and Equivariant Graph Networks, Haggai Maron, Heli Ben-Hamu, Nadav Shamir, Yaron Lipman, ICLR 2019, https://openreview.net/forum?id=Syx72jC9tm
> [4] Chen, Z., Villar, S., Chen, L. and Bruna, J., 2019. On the equivalence between graph isomorphism testing and function approximation with GNNs. arXiv preprint arXiv:1905.12560
> [5] Relational Pooling for Graph Representations, Ryan L. Murphy, Balasubramaniam Srinivasan, Vinayak Rao, Bruno Ribeiro, ICML 2019

---

> > ### Author Response · Authors · 2019-11-07
> > **Review answer (2)**
> >
> > 8. For the real-world datasets, we examined values of the parameter $k$ in $\{1,2,4,8,16\}$. Indeed, we omitted to clarify which value of $k$ we use in Table 1, as it was set as  a hyper-parametrization result and the best accuracy results were not achieved for the same value of $k$.
> > On real world datasets the number of colors did not have a large impact on the accuracy (as for other competitors such as RP-GIN), and we thus decided not to report these results to gain a little extra space. However, it is true that this is an important information for readers, and we will add a short discussion to the experimental section as well as the results of 0-CLIP (without any coloring), 1-CLIP and 16-CLIP to the appendix. Moreover, we will also add another experiment used by a recent related paper [4,5] to assess the quality of universal graph representations (see point 5.).
> >
> > We use bold in order to better clarify the statistically insignificant difference between the different algorithms used in the benchmark. We use the same standard statistical test than the one used for example in [2].
> >
> > 9. We took the standard definition from complexity theory [1]: a function f is said to be of exponential growth if log(f) is of polynomial growth.
> >
> > [1] Computational Complexity: A Modern Approach, Papadimitriou
> > [2] How powerful are graph neural networks, Xu et al., ICLR 2019, https://arxiv.org/pdf/1810.00826.pdf
> > [3] Invariant and Equivariant Graph Networks, Haggai Maron, Heli Ben-Hamu, Nadav Shamir, Yaron Lipman, ICLR 2019, https://openreview.net/forum?id=Syx72jC9tm
> > [4] Chen, Z., Villar, S., Chen, L. and Bruna, J., 2019. On the equivalence between graph isomorphism testing and function approximation with GNNs. arXiv preprint arXiv:1905.12560
> > [5] Relational Pooling for Graph Representations, Ryan L. Murphy, Balasubramaniam Srinivasan, Vinayak Rao, Bruno Ribeiro, ICML 2019

---

### Author Response · Authors · 2019-11-12
**Comment on the new submission**

Dear reviewers, as discussed in our answers to your comments, we have updated the submission with the following main changes:

1. Section 6 was replaced by a short comment in Section 5.

2. The main document is now 8 pages long.

3. We included new experiments on the Circular skip links problem of [1,2], and significantly outperform their algorithms (CLIP obtains a (max,min) accuracy over 20 runs of (98.7, 76.0) compared to (80, 10) for Ring-GNN and (53.3, 10) for RP-GIN).

4. We provide an ablation study on the benchmark graph classification datasets by providing the results of CLIP with the coloring mechanism (named 0-CLIP), as well as results for a varying number of colorings in the appendix.

[1] Relational Pooling for Graph Representations, Ryan L. Murphy, Balasubramaniam Srinivasan, Vinayak Rao, Bruno Ribeiro, ICML 2019

[2] Chen, Z., Villar, S., Chen, L. and Bruna, J., 2019. On the equivalence between graph isomorphism testing and function approximation with GNNs. NeurIPS 2019, arXiv preprint arXiv:1905.12560.

---

### Decision · Program_Chairs · 2019-12-19

**Decision:**

Reject

**Comment:**

This paper presents an extension of MPNN which leverages the random color augmentation to improve the representation power of MPNN. The experimental results shows the effectiveness of colorization. A majority of the reviewers were particularly concerned about lacking permutation invariance in the approach as well as the large variance issue in practice, and their opinion stays the same after the rebuttal. The reviewers unanimously expressed their concerns on the large variance issue during the discussion period. Overall, the reviewers believe that the authors has not addressed their concerns sufficiently.